# Current Advances in Photodynamic Therapy (PDT) and the Future Potential of PDT-Combinatorial Cancer Therapies

**DOI:** 10.3390/ijms25021023

**Published:** 2024-01-13

**Authors:** Niuska Alvarez, Ana Sevilla

**Affiliations:** 1Department of Cell Biology, Physiology and Immunology, Faculty of Biology, University of Barcelona, 08028 Barcelona, Spain; nalvarfu40@doct.ub.edu; 2Institute of Biomedicine, University of Barcelona (IBUB), 08036 Barcelona, Spain

**Keywords:** photodynamic therapy, reactive oxygen species (ROS), cancer, photosensitizer, phototoxicity, nanotechnology, quantum dots, immunotherapy

## Abstract

Photodynamic therapy (PDT) is a two-stage treatment that implies the use of light energy, oxygen, and light-activated compounds (photosensitizers) to elicit cancerous and precancerous cell death after light activation (phototoxicity). The biophysical, bioengineering aspects and its combinations with other strategies are highlighted in this review, both conceptually and as they are currently applied clinically. We further explore the recent advancements of PDT with the use of nanotechnology, including quantum dots as innovative photosensitizers or energy donors as well as the combination of PDT with radiotherapy and immunotherapy as future promising cancer treatments. Finally, we emphasize the potential significance of organoids as physiologically relevant models for PDT.

## 1. Introduction

As the global incidence of cancer continues to increase, there is a rising need for innovative and efficacious treatment approaches [1]. This demand extends to alternative therapies that can address the shortcomings of traditional treatments and improve patient outcomes. In this context, there has been a notable focus on treatments that harness the properties of molecular oxygen, leveraging its reactive states, which parallel those naturally employed by the body. This emphasis is especially prominent in the realm of photodynamic therapy [2].

It is widely recognized that cells naturally produce reactive oxygen species (ROS) to perform essential functions, such as cell signaling and defending against pathogens [3]. At physiological levels, ROS also play beneficial roles, including the regulation of gene expression, cell differentiation, and the maintenance of stem cells [4]. These endogenous ROS mainly originate from natural sources like mitochondrial respiration and various enzymes, including NADPH oxidases (NOXs), among other contributors. However, they can also be generated by external factors such as ultraviolet light (UV) and ionizing radiation (IR), drugs, and other substances like tobacco and alcohol [3]. When reactive oxygen species (ROS) levels exceed the physiological range, they can indeed become detrimental to cellular components, including nucleic acids, lipids, and proteins, causing cellular disruption and cell death through mechanisms like apoptosis, necrosis, and autophagy [3,4,5]. These adverse effects of ROS are also associated with processes related to aging, cancer, and neurodegenerative diseases [6]. Nonetheless, despite their potential for causing cellular damage, ROS are gaining recognition as valuable assets in cancer therapy.

One approach to exploit the potential of ROS for therapeutic purposes is through photodynamic therapy (PDT). PDT studies have shown promise in directing the harmful effects of ROS toward cancer cells by utilizing the body’s endogenous oxygen in combination with a photosensitizer (PS) and light at a specific wavelength [5,6,7]. Photodynamic therapy (PDT) is a minimally invasive method that has captured significant attention as an emerging tool for the treatment of cancer. Some studies have also reported its potential to offer several advantages when applied in combination with conventional therapies, such as chemotherapy and radiation [8,9].

The dynamics of PDT are mainly dependent on the electronic interactions between an excited photosensitizer compound (PS) and molecular oxygen. These further lead to energy transfer reactions, culminating in the generation of ROS, primarily singlet oxygen (^1^O_2_) [5]. PDT is gaining considerable recognition for its selectivity, relative ease of therapeutic application, high tolerability by patients, and efficacy in treating certain cancers, including inoperable tumors [10,11]. A significant advantage of this therapeutic approach is its ability to focus disruptive oxygenated reactions on a specific target site within the body, like tumors, thereby minimizing damage to the surrounding healthy tissue [11].

## 2. Photodynamic Therapy: Principles and Reaction Mechanisms

Photodynamic therapy (PDT) is a therapeutic approach that hinges on the combined effects of light, a photosensitizer (PS), and molecular oxygen. Initially, the patient is administered a PS, either through intravenous injection or topical application. Following this, there is a crucial period termed the ‘drug-light interval’. During this phase, the PS selectively accumulates in the target tissue, ensuring that the desired concentration is reached. Once the optimal distribution is achieved, light of a specific wavelength is applied to the target area, which subsequently results in the activation of the PS [5,12,13].

The selection of the wavelength is determined by the absorption spectrum unique to the chosen PS. For PDT applications, the chosen wavelengths typically fall within the range of 600 to 850 nanometers (nm) in the absorption spectrum [12,14]. This spectrum defines the range of light wavelengths that a PS can effectively absorb. Each wavelength within this spectrum contains the exact energy needed to activate the PS, thus enabling energy transitions. Activation requires that the absorbed light’s energy precisely matches the energy difference required for electrons to shift between their respective energy levels [13,15]. This concept is visually represented by the peaks in the absorption spectrum, which correspond to the energy differences between the electron energy states of the molecule [16].

Upon irradiation, and the subsequent absorption of the requisite energy, electrons within the PS are propelled to higher energy states. This facilitates the transition of the PS molecule from its stable ground state to an excited state (Figure 1) [13,17]. A ground state is characterized by electrons residing in the lowest energy level of an atom or molecule (n = 1), thereby maintaining minimal energy and ensuring stability due to their proximity to the nucleus and minimized interactions [18]. This shift from the ground to the excited state is critical, as it initiates photodynamic reactions and results in the transformation of the PS to a transient excited singlet state (^1^O_2_). In this state, the PS becomes electronically unstable and highly reactive, characterized by one electron occupying a higher energy level than its counterpart in the pair, thereby possessing surplus energy [13,19]. 

To clarify this concept further, typically, during this fleeting phase, the electrons within the PS, which ordinarily exist in pairs with opposite spins, undergo a process of unpairing (Figure 1). The unpairing specifically results from one electron in the pair absorbing energy and moving to a higher level, while the other remains at a lower energy level [13,18]. This altered electronic configuration, where electrons with opposite spins are temporarily unpaired and reside in distinct energy levels, contributes to the heightened reactivity of the singlet excited state [13,18]. The singlet excited state is transitory as electrons seek to return swiftly to a stable, paired configuration [13]. The energy imbalance and unpaired state of the electrons make the molecule more susceptible to engaging in chemical reactions [13,18]. Moreover, as the PS seeks to dissipate its excess energy and return to its ground state, it can engage in other mechanisms. Such mechanisms may include fluorescence, where the excess energy is emitted as light, or internal conversion, which involves the transformation of excess energy into heat [13,20].

Simultaneously, the singlet PS in its excited state can undergo intersystem crossing, where it transitions to a triplet state, the most efficient state for PDT applications [21]. The triplet state is characterized by having two unpaired electrons, like the singlet state. However, in the triplet state, these electrons have parallel spins, which makes it more stable compared to the singlet excited state [21]. In this brief yet reactive phase, the PS can efficiently transfer its excess energy to nearby molecules, like oxygen (O_2_) found in the blood, leading to the production of reactive oxygen species (ROS), notably singlet oxygen (^1^O_2_) [22]. This shift in the spin direction of an electron or intersystem crossing is a key process in PDT and typically occurs in a non-radiative manner since it takes place without the emission of a photon. The shift may be influenced by external factors, such as the presence of heavy atoms in the PS molecule, external magnetic fields, or the specific molecular environment. Additionally, intersystem crossing can also be significantly affected by a phenomenon known as spin–orbit coupling. This coupling is a natural quantum mechanical process involving the interaction between an electron’s spin and its orbital motion around the nucleus [21,23]. Furthermore, the longer-lived triplet state of the PS enables the direct transfer of energy to molecular oxygen (O_2_), resulting in the formation of singlet oxygen (^1^O_2_), the preferential byproduct resulting from the photochemical reaction [24,25]. 

PDT involves two main types of reactions: type I and type II. Both contribute to the therapeutic effects of PDT by inducing cell death through different processes, and they can occur simultaneously [5,26]. The balance between these reactions is influenced by several factors, including the nature of the PS, the substrate or target tissue, the local oxygen concentration, and the binding affinity of the sensitizer to the substrate [5,27]. Both Type I and Type II are characterized by the PS initially being in a singlet excited state due to photon energy transfer from light irradiation. In this state, the PS can then undergo intersystem crossing to progress to a more stable, long-lived, electronically excited triplet state. Each reaction has different product types and is primarily classified according to the mechanism involved [5,27]. 

In a Type I reaction, the triplet excited state of the PS directly interacts with a substrate, which could be a cell membrane or another molecule (Figure 2). This biochemical interaction can directly cause damage through two different processes: hydrogen atom abstraction or electron transfer reactions. Following this, highly reactive free radicals and radical ions can be produced from reactions with cellular components such as lipids from the lipid membrane and proteins to induce immediate cellular damage. Additionally, these free radicals can further engage with molecular oxygen (O_2_) to generate various reactive oxygen species (ROS), such as superoxide anions (O_2_^−•^), hydroxyl radicals (HO^•^), and hydrogen peroxide (H_2_O_2_). These ROS can cause subsequent oxidative damage to biological structures, ultimately leading to cell death [5,24,28,29]. However, the literature on hydrogen atom abstraction and energy transfer in the context of PDT is limited, as there is a scarcity of information available on these topics.

Castaño et al. are among the few to describe a parallel mechanism in Type I photodynamic therapy, where the triplet excited PS directly engages with molecular oxygen (Figure 2). This interaction prompts an electron transfer, resulting in superoxide anion production (O_2_^−•^). Superoxide itself, although reactive, is generally less damaging compared to other oxygen reactive species like the hydroxyl radical (HO^•^). Subsequently, superoxide is further processed into hydrogen peroxide (H_2_O_2_) and oxygen through the process of dismutation, which can occur spontaneously or enzymatically through the Superoxide Dismutase enzyme (SOD). The critical damage occurs when superoxide participates in the Fenton reaction react with metal ions like iron (Fe^3+^), causing their reduction (Fe^2+^). The reduced metals can then engage with H_2_O_2_ to generate highly reactive hydroxyl radicals. These hydroxyl radicals, capable of penetrating cells, initiate a series of reactions that severely harm cellular components by bonding with or extracting electrons from intracellular molecules, ultimately resulting in cell death [17].

The Type II reaction, on the other hand, predominantly occurs due to the direct transfer of energy from the triplet-excited PS to molecular oxygen (O_2_), which concludes in the formation of singlet oxygen (^1^O_2_) and the return of the PS to its ground state (Figure 2) [30]. The energy transfer specifically occurs because of collisions, or proximity interactions, between the excited-triplet-state PS and oxygen (O_2_). A PS molecule can produce a significant amount of singlet oxygen, typically ranging from 10^3^ to 10^5^ molecules, before it undergoes decay [30]. Singlet oxygen is highly reactive and can interact with numerous biological substrates, inducing oxidative damage, and subsequently, cell death. Notably, the Type II reaction is predominant during PDT, and singlet oxygen is identified as the primary cytotoxic agent responsible for the biological effects observed [30]. The molecules that are affected are determined by the location of the accumulated PS, and they can be either proteins, lipids, or DNA [31]. Moreover, the PS can undergo multiple cycles of energy transfer, repeatedly transitioning to and from its ground state without degradation [30]. Nevertheless, over time, it may experience a loss of activity, typically due to photobleaching [32].

Additionally, the literature describes another mechanism known as the Type III reaction in PDT. Unlike the Type II reaction, this pathway operates independently of oxygen, allowing a PS to directly interact with specific target biomolecules [33,34,35]. For the Type III mechanism to take effect, the PS itself should possess a specific targeting property toward proteins, nucleic acids, and other cellular molecules. Upon combination with a Type III PS, the biological target molecule can be directly and efficiently destroyed by the PS in its excited state following exposure to light. This mechanism offers a distinctive advantage by completely bypassing the ‘bottleneck’ of oxygen concentration that is often encountered in traditional PDT [33,34,35,36]. However, it is worth noting that only a limited number of PSs possessing this intrinsic selectivity feature have been identified thus far, as detailed in the following section of this review. Further research is strongly encouraged to identify and develop more promising Type III reaction molecules. These molecules have the potential to overcome the limitations associated with oxygen availability in current PSs, thus expanding the effectiveness of PDT [35].

## 3. Evolution of Photosensitizer Design

As ongoing clinical investigations continue to generate evidence in the field of PDT, a wide variety of PSs have emerged over the years [36]. In cancer therapy, PSs have been categorized according to their usage and effectivity for specific cancer types [36,37]. Most of the PSs approved for cancer treatment are derived from the tetrapyrrole molecule, resembling the structure of a component of hemoglobin known as the protoporphyrin group [37]. These compounds belong to the first generation of PSs and include the hematoporphyrin derivative (HpD) and photofrin II, a purified form of HpD [38]. They have found applications in treating lung, colorectal, brain, and breast cancers. However, the first-generation PSs had several limitations due to their complex molecular structures, synthesis challenges, low quantum yields, and hydrophobic nature. These factors, combined with limited selectivity, significantly impeded their ability to penetrate tissues [39]. To overcome these constraints, efforts were made to modify existing molecules by introducing chemical groups into their tetrapyrrole ring. While these modifications improved the solubility to some extent, not all issues were resolved [40]. 

The significant breakthrough in the field of PSs came with the introduction of the second generation. Phthalocyanines and chlorins, which are part of this generation, proved to be more efficient, offering increased selectivity and improved tissue penetration, especially in the near-infrared spectrum, making them valuable for cancer treatment [38,39,40]. An example of a widely recognized second-generation PS is 5-aminolevulinic acid (5-ALA). When metabolized, it transforms into protoporphyrin IX (PPIX), which functions as a PS [39]. PpIX naturally occurs in all living cells and is used for the synthesis of heme groups, essential iron-containing compounds within hemoglobin [41]. 

Administering 5-ALA to oncology patients results in a higher PpIX aggregation in cancer cells compared to normal cells [39]. The selective accumulation of PpIX in tumor cells has been attributed to the altered metabolism of these cells, which is distinct from that of normal cells. Tumor cells exhibit a higher reliance on glycolysis for energy production, a phenomenon known as the Warburg effect, rather than oxidative phosphorylation. Consequently, tumor cells do not efficiently produce heme, which in turn results in the accumulation of PpIX within the tumor cells [42]. 

An advantage of PpIX over other porphyrin-based photosensitizers is its relatively short elimination time (typically within 28 to 48 h), reducing the risk of long-term photosensitivity [39]. However, individuals with certain porphyrias, a genetic disorder, can experience PpIX accumulation, which potentially leads to liver failure [43]. Boronated porphyrins (BOPPs) are another class of second-generation PSs. When administered to a patient, BOPPs enable a dual treatment approach, combining both PDT and Boron Neutron Capture Therapy (BNCT) to target tumors, especially in the brain [44]. This combination therapy takes advantage of boron’s natural affinity for cancerous tissue, providing an additional layer of treatment. However, BNCT’s clinical applications are evolving, and its availability varies across healthcare settings due to ongoing research and development initiatives [44].

Recent efforts have led to significant advances in the development of a third generation of PSs. These novel compounds have been built upon well-known PS molecules like porphyrins, chlorins, and phthalocyanines. They are ingeniously combined with specific proteins, amino acids, antibodies, or carbohydrates, conferring them the ability to precisely target cellular components and disrupt vital cellular functions [45]. This new class of PSs often employs advanced delivery systems, including attachment or encapsulation with nanoparticles [45]. Such innovations hold the potential to enhance PS specificity, which can result in a higher accumulation at the tumor site and reduced systemic toxicity [45].

In parallel with the development of third-generation PSs, the incorporation of nanozymes has also emerged as a noteworthy advancement in the field [45,46]. Nanozymes are nanomaterials with enzyme-like properties capable of catalyzing various chemical reactions [45]. In the context of PDT, they have been shown to regulate tumor oxygen deficiency as well as intensify the generation of ROS at the localized target site, ultimately leading to the more effective and rapid destruction of cancer cells [45,46]. Furthermore, ongoing research in nanotechnology has highlighted its significant potential in PDT applications, as it enhances drug delivery precision, targetability, safety, and overall treatment efficacy in cancer therapy, some of which are discussed in the following section [47].

As we can observe, most of the advances have been oriented toward enhancing the strength of PDT as a therapeutic approach in cancer, which relies on both light and PS accumulation. Thanks to this bifunctionality, PDT has been oriented to be more target-specific, causing less damage to healthy tissues within the body [37].

Accumulating the knowledge from the different generations of PSs, an optimal PS molecule should be selected based on specific criteria, with the first parameter being its absorption spectrum, ideally falling within the range of 600 and 800 mm [34]. Longer wavelengths, such as red and infrared, are preferred because they can penetrate deeper into the target tissue, as they are less likely to be absorbed and scattered by tissue components [34,37]. Precise control of the wavelength is crucial, as exceeding 800 nm can result in insufficient energy for exciting oxygen to a singlet state, rendering the photosensitivity reaction nonviable [34,37].

The selected PS should also be able to attain an appropriate quantum yield of its triplet excited form. In other words, an optimal number of PS molecules must reach a triplet excited state, following irradiation, to enable maximal interactions with oxygen, leading to the subsequent generation of a substantial quantity of ROS [34]. In terms of toxicity, the PS should possess the ability to clear non-target tissues relatively quickly and must remain non-reactive in the absence of light [34,37]. While it is advisable to have a long therapeutic window, allowing the PS to disperse from normal tissue and accumulate in the tumor, some studies now suggest that applying light shortly after PS administration might enhance effectiveness. This is because, during this early stage, a significant amount of the PSs may remain in the tumor blood vessels, potentially resulting in significant vascular damage and better treatment outcomes [37,48].

In further considering the selection of an ideal PS, it is essential to factor in the potential benefits of photobleaching that refers to the process in which a PS becomes damaged or destroyed when exposed to light [37]. This process has also been found to be advantageous in PDT treatment by contributing to the control of light dosage. Photobleaching can help to prevent over-treatment by limiting the continued activation of the PS, ensuring that the therapeutic effect is precisely targeted [37,49].

## 4. Mechanisms of Cell Death and Immunogenic Responses in Photodynamic Therapy

As it has been previously remarked, the application of PDT primarily yields reactive oxygen species (ROS), predominantly singlet oxygen (^1^O_2_). The interaction between ROS and vital biomolecules leads to the eradication of cancer cells, the disruption of tumor vasculature, and the activation of immune cellular responses. Consequently, these processes collectively culminate in the destruction of the tumoral cells [43].

Currently, we have knowledge of the mechanisms of cell death triggered by the cell following the photodynamic reaction. Among these, apoptosis, necrosis, and autophagy seem to be the prevailing pathways [50]. However, the specific cytotoxic pathway marking the cell’s faith can vary depending on the location where the PS accumulates after administration [50,51]. For instance, it has been suggested that the accumulation of the PS in the cell’s plasma membrane may potentially lead to necrosis [50,51,52]. While the precise steps governing the mechanisms of necrosis following PDT are still under investigation, it has been established that PDT disrupts the plasma membrane, resulting in the abrupt release of ATP [50,51].

Conversely, when the PS accumulates within the mitochondria or organelles, it generally initiates apoptosis, which involves the activation of effector caspases [50,51]. However, another apoptotic mechanism has also been identified, involving the mitochondrial apoptosis-inducing factor (AIF) instead of caspases [53]. Apoptosis induced by PDT represents a key pathway of cell death [54]. It serves as an alternate route that can be activated following PS administration and subsequent light exposure. This apoptotic process operates through two distinct pathways: the intrinsic pathway centered on mitochondria and the extrinsic pathway initiated by external factors at the plasma membrane [50].

Intrinsic apoptosis starts when the PS accumulates within mitochondria, triggering ROS generation in this organelle. These ROS then disrupt the mitochondrial membrane, releasing cytochrome C into the cytoplasm of the cell. Cytochrome C initiates a series of events that activate caspases, which are responsible for initiating cell destruction [55]. Typically, phagocytic cells are responsible for removing the deceased cells. However, it is essential to highlight that the role of ROS in disrupting the mitochondrial membrane and releasing cytochrome C is a multifaceted process with numerous intricate steps and regulatory factors involved [55]. 

So far, recent evidence has shown that the interplays between apoptosis and necrosis have also been found to rely on light fluence. While necrosis tends to be the most common form of cell death under strong photosensitizing conditions, apoptosis appears to be prevalent when light fluence and PS concentrations are reduced [56,57].

On the other hand, other studies have suggested that autophagy can promote a pro-survival effect as an independent mechanism of cell death, separate from apoptosis [50]. In this line, recent studies have described that even though autophagic proteins have been shown to be upregulated by the hypericin-induced combination treatment of photobiomodulation and PDT in human dermal fibroblasts (HDFs), autophagosome degradation was inhibited in this HDF. Therefore, autophagy seems to have a pro-survival effect in HDF under this treatment but not in U87 glioblastoma cells, thus giving a selective cell death toxicity [58].

Moreover, depending on the specific PS and treatment protocol employed, PDT has been associated with immunogenic cell death (ICD), another form of tumor cell death capable of eliciting adaptive immune responses linked to the recognition of cancer cells. This can occur through the release of different signaling molecules that serve as stimuli for the immune system, facilitating the identification of malignant cells [53,59]. Multiple studies have correlated ICD with the release of Damage-Associated Molecular Pattern (DAMP) molecules following treatment with PDT [60]. DAMPs are molecules that are normally sequestered within live cells (where they perform predominantly non-immunological functions), but acquire immunomodulatory activities once secreted or surface-exposed by dying or stressed/damaged cells [61]. These danger signals in the company of cancer cell constituents and antigens cause the maturation of dendritic cells (DCs), which ultimately ‘cross-prime’ and activate anti-tumorigenic CD4^+^/CD8^+^ T-cell immunity [61,62]. Numerous studies, as seen in the following sections, have illuminated the potential of PDT, not only for the destruction of tumors and their vasculature but also for harnessing ICD to amplify the body’s immune reaction against cancer. Furthermore, this approach offers the advantage of extending the treatment to tumors that may be less susceptible to light penetration [50].

## 5. Addressing Challenges in PDT: Some Promising Solutions

While PDT offers several advantages, it is essential to acknowledge the limitations it presents [50]. Firstly, PDT’s efficacy is influenced by the availability of oxygen, making it more effective in well-oxygenated tumor regions and less efficient in areas with low oxygen levels. Additional challenges include the distribution of photosensitizers (PSs) when administered intravenously, the incomplete delivery of light to the tumor site, issues related to target selectivity, incomplete tumor eradication, and partial disruption of the tumor vasculature [50]. Furthermore, the risk of damage to healthy tissues due to high light intensity when using conventional PSs presents an added challenge [63]. In the following section, we explore some of the recent advancements in PDT and discuss potential alternative solutions to some of the challenges presented in select studies, to promote further research in this field.

### 5.1. Elevating Photodynamic Therapy’s Efficacy in Hypoxic Tumors

Solid tumors present a significant hurdle for PDT. The primary mechanism of action for the first two generations of PSs relies entirely on the destruction of molecules through the influence of singlet oxygen. However, the tumor microenvironment (TME) presents a low oxygen (O_2_) supply caused by abnormal vascularization in neoplastic tissues and a high O_2_ consumption induced by the rapid proliferation of tumor cells. Certainly, when oxygen is in short supply, PDT’s therapeutic effectiveness is greatly reduced [50]. To address this problem, researchers have been developing advanced nanoplatforms and strategies to enhance the therapeutic effect of PDT in tumor treatment. Such strategies include delivering O_2_ to the tumors, the in situ generation of O_2_, and decreasing the O_2_ consumption during PDT by the design of Type I photosensitizers. As O_2_ carriers, some nanomaterials have been developed such as hemoglobin (Hb), red blood cells (RBCs), perfluorocarbons (PFCs), and metal–organic frameworks (MOFs) [64,65]. Alternatively, in situ O_2_ generation through bio-chemical reactions between the components in drug carriers and the endogenous substances in tumor tissues have also been explored. Among them, the use of H_2_O_2_ as oxygen-containing molecules to relieve tumor hypoxia has been widely confirmed [66]. To date, catalase (CAT) and CAT-like enzymes are applied to catalyze the decomposition of H_2_O_2_ into water (H_2_O) and O_2_, to relieve tumor hypoxia [67,68,69,70,71]. However, in this case, O_2_ production is also limited due to the limited H_2_O_2_ content in the tumor.

Earlier, we explored Type III reactions as a supplementary potential mechanism in PDT. Indeed, PSs capable of directly interacting with biomolecules within cancer cells, irrespective of a low-oxygen environment, have been proven to be especially valuable in addressing hypoxic tumors [72]. As an example, a study introduced a novel PS known as NBEX (X = S, Se, Te), which belongs to the class of N-heterocyclic dyes. NBEX has the unique capability to selectively bind to RNA within the cellular environment without interference from other components. As a remarkable feature, these newly designed dyes can autonomously assemble into nanoparticles. When exposed to light irradiation, they harness their excitation energy and directly transfer it to the RNA present in cancer cells. In vivo experiments have confirmed the accumulation of NBEX in tumors, where it infiltrates the cells and binds to RNA. This process ultimately leads to the destruction of cancer cells, making it a valuable approach for treating tumors in hypoxic conditions [35].

In an alternative approach, the PS can be strategically engineered to produce its oxygen supply, rendering it self-reliant for its therapeutic function. As a case in point, in a notable investigation, gold nanorods were specifically designed to exhibit near-infrared light absorption properties and were further customized to incorporate endoperoxides into their structure. When these nanorods are exposed to light, they undergo a heating reaction, resulting in the generation of singlet oxygen from the endoperoxides. This critical step played a pivotal role in disrupting molecules in the surrounding vicinity, ultimately affecting the targeted tumor cells [73].

Finally, a recent synergistically combined therapy of PDT and other therapeutic methods such as photothermal therapy (PTT), immunotherapy, chemotherapy, and gas therapy are accounted for by addressing the challenging problems of mono-PDT in hypoxic environments, including tumor resistance, proliferation, and metastasis [74]. 

PTT is a unique non-invasive phototherapy method that destroys tumor tissues by converting light energy into heat energy using materials with an ideal photothermal conversion efficiency without the need of O_2_ [75]. One of the advantages of this combined therapy is that mild hyperthermia can increase the membrane permeability to enhance the tumor cell uptake of PS-loaded nanocarriers; therefore, as the intracellular PS concentration is increased, the intracellular ROS concentration also increases, enhancing the PDT efficacy. Furthermore, PTT, like phototherapy and PDT, not only destroys tumor cells but also elicits immunogenic cell death (ICD) and can effectively facilitate the inhibition of distant tumor metastasis. This ICD effect of synergistic PDT/PTT is much stronger than PDT or PTT alone [76,77]. Therefore, this synergistic approach can double the therapeutic effect.

In the same manner, PDT combined with immunosuppressants or immunoadjuvants can also achieve superior antitumor ability in a hypoxic environment. Tumor cells can downregulate immune checkpoints to evade immunity by interfering with certain signaling pathways to regulate T-cell activity. Ctla-4 (cytotoxic T-lymphocyte-associated antigen 4) and PD-1/PD-L1 (programmed death receptor-1/programmed death ligand-1) immune checkpoint inhibitors are currently available for clinical use [78,79]. To better promote the efficacy of PDT in a hypoxic environment, Lan et al. reported a nano-MOF (Fe-TBP) consisting of Fe_3_O clusters and 5,10,15,20-tetra (p-benzoate) porphyrin (TBP) that, when irradiated under hypoxic conditions, catalyzed a cascade reaction in which intracellular H_2_O_2_ was decomposed by the Fe_3_O clusters to produce O_2_ through a Fenton-like reaction, whereas the generated O_2_ was converted to cytotoxic singlet oxygen (^1^O_2_) by photoexcited porphyrins [80]. Meanwhile, Fe-TBP combined with anti-programmed death-ligand 1 (α-PD-L1) induces a significant cytotoxicity amplification as it increases CD4^+^ and CD8^+^T circulating cells, causing a 90% ablation of tumoral cells, leading to the regression of both treated primary tumors and untreated distant tumors. More recently, Sun et al. have reported a porphyrin-based metal–organic framework combined with hyaluronate-modified CaO_2_ nanoparticles (PCN-224-CaO_2_-HA) to target and enhance PDT efficacy. CaO_2_ reacts with H_2_O or a weak acid to produce O_2_, overcoming the hypoxia problem. Hyaluronate protects CaO_2_ and specifically targets the CD44 receptor, which is highly expressed on tumor cell membranes, achieving an enhanced targeted therapy efficacy in vitro and in vivo [81]. In recent years, PDT combined with chemotherapy has progressed in the preclinical and clinical stages of tumor therapy. The chemotherapeutics may improve the sensitivity of cancerous cells to ROS, while ROS in turn suppresses the drug-efflux activity of cells and enhances the cellular uptake of drugs [82]. Huang et al. reported a hypoxia-activated system for PDT combined with chemotherapy, named AQ4N@CPC-FA [83]. The combination of the two treatments improves the efficacy of PDT in the hypoxic tumor, reduces drug resistance, and enhances the efficacy of cancer eradication. Supramolecular micelles (CPC) were derived from a polyethylene glycol (PEG) system dually tagged with hydrophilic cucurbit[7]uril (CB[7]) and hydrophobic Chlorin e6 (Ce6), respectively, on each end, for synergistic antitumor therapy via the PDT of Ce6 and the chemotherapy of a hypoxia-responsive prodrug, banoxantrone (AQ4N), loaded into the cavity of CB[7]. The binding of CPC with folate (FA) targeted excess FA receptors on the surface of tumor cells. Under the irradiation of a laser, Ce6 produced ^1^O_2_ for PDT and exacerbated tumor hypoxia, thereby activating AQ4N to be converted into chemotherapy drug AQ4.

Moreover, using hypoxia-sensitive compounds such as azomycine analogs (2-nitroimidazole, 4-nitrobenzyl, etc.), azobenzene derivatives, quinones, aliphatic N-oxides, aromatic N oxides, and transition metals, researchers have successfully designed vehicles carrying PSs and anticancer drugs for hypoxia-activated drug release [84,85]. One recent example is the combination of an albumin-based nanoplatform co-delivering IR780, the NLG919 dimer, and a hypoxia-activated prodrug tirapazamine (TPZ) as the dual enhancer for synergistic cancer therapy. Under NIR irradiation, IR780 generates ^1^O_2_ for PDT, which simultaneously cleaves the ROS-sensitive linker for triggered TPZ release, and activates its chemotherapy via exacerbated tumor hypoxia. Surprisingly, it has been observed that TPZ-mediated chemotherapy boosts PDT-induced tumor ICD to evoke a stronger antitumor immunity, including the development of tumor-specific cytotoxic T lymphocytes (CTLs).

Finally, gas therapy (GT) has recently attracted more attention as a novel cancer treatment with good therapeutic effects and minor side-effects [86,87]. Studies have shown that gases such as nitric oxide (NO), hydrogen sulfide (H_2_S), sulfur dioxide (SO_2_), carbon monoxide (CO), and hydrogen have the potential for cancer therapy [88,89,90,91]. Compared to other synthetic oncology drugs, these gases have as advantages small molecular weights and, because of that, they can easily diffuse deep into tumor stroma and across biofilms without active transport mechanisms. They are used either to kill tumor cells directly or to make tumor cells more sensitive to current anticancer methods such as PDT, chemotherapy, and radiation therapy [92].

NO can directly mediate tumor cell death through mitochondrial/DNA damage, the inhibition of DNA repair, and cell respiration [93,94]. Moreover, NO is also a PDT sensitizer under hypoxic conditions, as it reacts with ROS, producing highly toxic active nitrogen species, such as nitrous oxide and peroxynitrite anion [92]. Wan et al. designed a tumor-specific ROS-responsive nanoplatform capable of the combination of nitric oxide (NO)-based gas therapy and sensitized PDT. This nanoplatform consisted of porous porphyrin MOF PCN-224 as a carrier to carry l-arginine (L-Arg), containing NO donor L-Arg that was concurrently coated with a cancer cell membrane (L-Arg@PCN@Mem). This platform, L-Arg@PCN@Mem, under near-infrared light (NIR) irradiation, produces plenty ROS directly for PDT therapy, while part of the ROS take the role of oxidative stress to convert L-Arg into NO for combined gas therapy, enhancing the sensitization to PDT under hypoxic conditions [95]. Considering the short half-life and gaseous state of NO, key factors like accurate release along with spatial, temporal, as well as dose control should be taken into consideration when designing synergetic GT/PDT.

Another approach using SO_2_-mediated GT combined with PDT to treat tumors was also explored by Wang et al. where an organic activatable PS, CyI-DNBS, bearing 2,4-dinitrobenzenesulfonate (DNBS), acts as the cage group [96]. Notably, CyI-DNBS is uptaken by cancer cells after which the cage group is selectively removed by intracellular glutation, resulting in the generation of SO_2_ for GT. The reaction also releases the activated PS, CyI-OH, that can produce singlet oxygen (^1^O_2_) under red light irradiation. Therefore, CyI-DNBS targets cancer cells for both photodynamic and SO_2_ gas therapy treatments. Although gas therapy has shown good results, the design of the nanocarriers with controllable release still needs to achieve accurate gas delivery and reduce the risk of gas toxicity as H_2_S, SO_2_, and CO can cause damage to normal tissues after systematic administration.

### 5.2. X-ray-Induced Photodynamic Therapy (X-PDT) for Improved Tissue Penetration and ROS Generation

One of the additional challenges hampering the efficacy of PDT is the limited depth to which light can penetrate certain tumors [97]. PDT relies on visible light, which emits shorter wavelengths that are prone to absorption and scattering by biological tissues. As light travels deeper into the body, its intensity diminishes, compromising the activation of the photosensitizing agent [98]. To address these challenges related to light penetration, recent studies have explored alternative energy sources with better tissue-penetrating capabilities. This approach combines PDT with radiotherapy, known as X-ray-induced photodynamic therapy (X-PDT) [97,99]. Radiotherapy (RT) remains one of the most used cancer treatment methods, administered to over 50% of cancer patients as part of their treatment regimen [100]. However, RT faces various obstacles that limit its effectiveness, including unresponsive cancer cells, the development of resistance to radiotherapy by cancer cells, and the potential for collateral damage to healthy tissues, leading to poor prognosis [101,102]. X-ray irradiation has been found to overcome the constraints of PDT associated with limited light penetration into the tumor region [97,103]. X-rays possess greater tissue-penetrating capabilities due to their higher energy and longer wavelengths, allowing them to reach the PS and activate it [99]. Several studies have demonstrated the potential of X-PDT to address the restrictions linked to radiotherapy when used in isolation. These studies suggest that X-PDT can work more effectively than when PDT and radiotherapy are applied separately [104].

In X-ray PDT, two primary components are typically used: nanoscintillators and PSs. Nanoscintillators in X-PDT are materials that possess the capability of absorbing X-ray energy and converting it into visible or NIR light, thereby exciting neighboring PSs and leading to the generation of ROS. This dual mechanism has demonstrated an improvement in the efficacy of X-PDT to reach and treat deep-seated tumors [80,105,106]. Utilizing scintillators to activate PSs in X-PDT seems to be more prevalent in the literature than direct X-ray activation of the PS. This preference stems from the limited success of X-rays in effectively activating traditional PSs in previous attempts, highlighting the more reliable efficacy of scintillators in this role [107].

However, in some instances, PSs can be directly activated by X-rays. In a recent study, copper–cysteamine complex (Cu-Cy) nanoparticles were introduced as a new type of X-ray-activated PS for the treatment of cancer. These nanoparticles directly produced singlet oxygen upon X-ray exposure. In both in vitro and in vivo studies involving human breast cancer cells (MCF-7), Cu-Cy nanoparticles demonstrated significant effectiveness in cell destruction with X-ray activation [107]. Additionally, in certain implementations of X-rayPDT, the scintillators and PSs are introduced separately, rather than being conjugated or combined. Typically, the scintillators are conjugated or chemically bonded to the PSs to ensure efficient energy transfer from the scintillator to the PS [105]. An example of such conjugations are the MC540-SAO: Eu@mSiO_2_ nanoparticles, which have been demonstrated to significantly enhance the effectiveness of radiotherapy when combined with PDT. These nanoparticles act as scintillators, converting X-ray energy into visible light, and as PSs, producing singlet oxygen upon X-ray exposure to induce cell death. This approach was shown to be particularly effective against cells resistant to conventional radiotherapy [80,108].

Furthermore, Sours et al. classified nanoparticles in X-PDT into four main groups based on their scintillator material: rare earth elements, metals, silicon, and quantum dots (Figure 3). Scintillators and PSs are often combined within these nanoparticles, either delivered independently or optimized together for enhanced effectiveness. Rare earth elements, like doped nanoscintillators, were among the first to be studied, frequently paired with organic PSs for efficient ^1^O_2_ production [105,109].

However, a significant drawback has been the predominance of inorganic scintillators in X-PDT applications, potentially causing issues related to toxicity [86]. In a recent breakthrough, a novel organic aggregation-induced emission (AI) nano-scintillator called TBDCR NPs has been introduced, offering a promising solution. This study highlighted its capacity to efficiently produce both Type I and Type II ROS when subjected to X-ray irradiation. TBDCR was intentionally engineered with the incorporation of heteroatoms into its molecular structure to enhance its ability to capture X-ray energy effectively. Moreover, its composition was fine tuned to significantly enhance the generation of ROS, with particular emphasis on hydroxyl radicals (HO^•−^), when exposed to X-ray irradiation. Remarkably, TBDCR exhibited aggregation-induced emission (AI), meaning that it emits fluorescence when its molecules aggregate. These distinctive characteristics position the use of organic nano-scintillators as valuable materials for X-ray-induced PDT, enabling the efficient utilization of X-ray energy and ROS production, even in oxygen-depleted conditions and with fewer side-effects [86].

### 5.3. Enhancing Light Emission and Intensity with Quantum Dots in Photodynamic Therapy

As advancements are made in the development of biocompatible and non-toxic molecules for PDT, the exploration of nanotechnology applications is underway. These studies examine the use of nanotechnology both independently and in conjunction with PSs for cancer treatment [63]. Recent initial investigations have indicated the potential benefits of utilizing quantum dots (QDs) for PDT. QDs are synthetic nanomaterials crafted from semiconductor elements, often composed of metals like cadmium selenide (CdSe), cadmium sulfide (CdS), cadmium telluride (CdTe), and lead sulfide (PbS). They can also be created from silicon, carbon, or graphene [110]. They represent part of the newest generation of nanomaterials and are essentially clusters of semiconductor atoms meticulously arranged in a crystalline structure a few nanometers in size [63]. These semiconductor particles have optical and electronic properties that differ from larger particles due to quantum mechanics. When the QDs are illuminated by UV light, an electron in the quantum dot can be excited to a state of higher energy. For a semiconducting quantum dot, this process corresponds to the transition of an electron from the valence band to the conductance band [111]. Then, the excited electron can drop back into the valence band, releasing its energy by the emission of light, and the color of that light depends on the energy difference between the conductance band and the valence band [112]

The discovery of these QDs offers certain advantages distinct from the previous traditional approaches [63]. QDs present size-dependent properties. Typically, smaller quantum dots demand higher energy levels to stimulate the electrons within their valence band. While conventional PSs do emit some light in the near-infrared spectrum, their primary emission falls mostly within the visible range. Consequently, not all of their emitted light rests within the near-infrared range, potentially resulting in some loss for specific applications that necessitate a specific wavelength in the near-infrared spectrum, such as tissue penetration [63]. In contrast, quantum dots can be meticulously engineered to regulate and enhance light emission, especially in the near-infrared spectrum, to enhance tissue penetration at the deep-seated tumors [113]. This capability facilitates the use of low-intensity light, a preference over higher-intensity light due to its reduced risk of nearby tissue damage and its ability to penetrate deeper into targeted tissues. This makes quantum dots a promising advancement for applications that require efficient near-infrared light emission [63].

In addition, what truly distinguishes them from ordinary materials is their remarkable characteristic known as electron confinement [63]. Within the confines of quantum dots, electrons are locked into specific energy levels due to their limited spatial capacity, arising from their diminutive size (typically ranging from 1 to 10 nanometers). This unique feature results in quantized energy levels [114]. Unlike materials where electrons can occupy a range of energy values, which may result in less efficient energy transfer processes, the quantized energy levels of QDs ensure precise energy jumps and focused outcomes in applications like PDT [63,114]. The electron confinement property offers precise control over their behavior and can be tailored for diverse applications. Thanks to these distinctive attributes, quantum dots boast tunable energy levels that can be finely adjusted to emit light at precise wavelengths and result in the generation of singlet oxygen through triplet energy transfer [63].

Additionally, the generation of ROS by quantum dots (QDs) is a distinctive process that sets them apart from conventional PSs. This process is primarily driven by the QDs’ unique electronic structure and surface states. When these QDs are exposed to light, they can efficiently transfer energy or electrons to nearby oxygen molecules, leading to the formation of various ROS types like singlet oxygen, superoxide anions, and hydroxyl radicals. The diversity in ROS production is linked to the QDs’ size, shape, and surface chemistry, which influence the potential pathways in their excited state.

QDs differ from traditional PSs in their ability to absorb a broader spectrum of light wavelengths. This broad absorption is attributed to their tunable electronic structures, allowing more effective energy transfers to oxygen molecules and the generation of diverse ROS types. In contrast, conventional PSs usually have limited absorption ranges and may produce fewer ROS types. Moreover, QDs can be engineered for specific targeting, enhancing their effectiveness, particularly in PDT applications [115,116].

A recent advancement in PDT involves the use of graphene quantum dots (GQDs). These GQDs have solved common limitations found in current PDT agents, such as low singlet oxygen quantum yields and poor biocompatibility. Utilizing a multistate sensitization (MSS) process, GQDs achieve a high quantum yield of around 1.3, significantly surpassing traditional agents. This is facilitated by their broad absorption in UV and visible light spectra and their strong deep-red emission. In GQDs, ROS generation can occur from both the excited singlet (S1) and triplet (T1) states, a mechanism different from traditional agents where singlet oxygen is typically generated through energy transfer from the excited triplet state. The high quantum yield in GQDs results from energy transfers from both the S1 and T1 states to molecular oxygen [115]. GQDs represent a breakthrough in PDT agents, offering a novel mechanism for ROS generation with high quantum yield and biocompatibility. However, since the use of QDs in PDT for cancer is relatively new, further studies are needed to elucidate the detailed mechanisms of their pharmacodynamics and pharmacokinetics, as well as in vitro studies shedding light on their cytotoxic effects [63]. So far, concerns have arisen regarding the use of heavy metals as a treatment. For example, a long-term investigation found that cadmium (Cd) from quantum dots can accumulate in the kidneys and liver [117]. This toxicity has been attributed to the leaching of the metal, which is the release of heavy metal ions, such as cadmium ions from the CdSe quantum dots, into the surrounding environment. This, in turn, leads to damage to healthy cells [63]. On the contrary, better outcomes in terms of toxicity and biocompatibility have been obtained with the use of carbon quantum dots instead of metals [63,118]. Several studies have reported the biosafety of carbon-based quantum dots for therapeutic purposes, potentially indicating their suitability for the development of new biomedical applications [119]. Recent human in vitro studies suggest that carbon quantum dots indeed exhibit low to no cytotoxicity, as evidenced by a higher number of viable cells following treatment with carbon QDs when compared to metal QDs [120]. Additionally, graphene quantum dots have also been reported as non-toxic and biocompatible for human cells after 24 h of exposure at studied concentrations [63,121].

Overall, quantum dots are gaining interest in the PDT field due to their ability to overcome the need for traditional PSs by potentially producing singlet oxygen [63]. When conjugated with a PS, they have also been suggested to enhance the photoluminescence of traditional PSs, thereby increasing the efficiency of the photosensitization process and leading to improved therapeutic outcomes [63]. An additional potential benefit could arise from attaching specific biomolecules to improve specificity toward the therapeutic target. However, new in vitro and clinical studies investigating the use of quantum dots are needed to shed more light on their safety, absorption, metabolism, excretion, and physiological responses [63].

### 5.4. Harnessing Immunological Strategies in Photodynamic Therapy

Among the challenges discussed earlier, one frequently encountered issue is the inability of PDT to completely eradicate tumors and prevent their recurrence and metastasis [122]. Several studies have revealed that PDT, under specific conditions, can induce immunogenic cell death (ICD) [122,123]. This response is initiated by the release of DAMPs and tumor-specific antigens, potentially leading to the activation of an adaptive immune response against any remaining cancer cells while also building immune memory [122,124]. In recognition of the potential of immunological PDT in combating tumors through ICD, emerging studies are increasingly harnessing the immunological reactions resulting from cancer cells treated with PDT. This strategy aims to facilitate the infiltration of immune cells into tumors and enhance the body’s antitumor response [122].

Immunological approaches can enhance treatment specificity to minimize harm to healthy tissues. While PDT has a notable selectivity for cancer cells, this can vary depending on the PS used, as there are challenges in achieving optimal tumor specificity and PS accumulation [62]. To address these issues, ongoing research is exploring the integration of immunological strategies, such as antibody conjugates, to enhance the precision of PDT and minimize damage to healthy tissues [62].

For instance, cetuximab, a monoclonal antibody (mAb) that binds to the epidermal growth factor receptor (EGFR), which is typically overexpressed by several cancer types, has been recently developed into a version conjugated to IR700, a photosensitizer that absorbs light at 690 nm. Upon irradiation, this combination has demonstrated the ability to destroy cancer cells expressing EGFR and generate an immune response [62,125]. While these examples showcase instances where PS design can be improved in terms of specificity with the help of immunotherapy, the possibilities for conjugation with IR700 or other PS combinations are vast and remain an area of active research [62].

Additionally, ongoing investigations are revealing promising synergies between immunotherapy and PDT, further amplifying the potential of this treatment approach. For example, a pre-clinical study demonstrated that Bremachlorin-based PDT in combination with immunotherapy (IPDT) can elicit an even stronger immune response, resulting in a significant tumor eradication [122]. In this study, PDT alone triggered a CD8^+^ T-cell antitumor response. Remarkably, the combination of PDT with synthetic long peptides containing epitopes from tumor antigens proved to be more efficient for tumor eradication than PDT alone. This enhanced efficacy is attributed to a strengthened CD8^+^ T-cell response, which also seems to confer protection against tumor recurrence [122,126].

In many cases, tumors create an immunosuppressive environment as a defensive strategy. However, specific organic PSs have shown the ability to trigger processes like ICD or macrophage polarization, making the tumor environment more accessible to the immune system. The ideal approach is one that not only destroys the tumor but also enhances the tumor microenvironment’s immune responsiveness to achieve a favorable antitumor response [122]. One promising strategy involves enhancing the therapeutic potential of commonly used PSs by incorporating additional functionalities. For instance, a study explored the combination of chlorin e6 (Ce) with plasmid DNA encoding the catalase gene (pDNA-cat) to enhance its accumulation within murine 4T1 tumors. This innovative nanomedicine, named SPM-P/C, effectively triggered a robust immune response. It facilitated the maturation of dendritic cells (DCs) and the infiltration of T-cells following PDT [122,127].

Additionally, an innovative platform combined the properties of the PS methylene blue with an immune gene plasmid encoding *IL12* gene (p*IL12*). To effectively deliver both components, they were encapsulated within magnetic mesoporous silica nanoparticles, which acted as carriers. The nanoparticles were subsequently taken up by mesenchymal stem cells (MSCs) due to their inherent magnetic properties. This integrated bio-platform, known as MB/IL12-MSCs, not only successfully reached the tumor but also infiltrated it. Consequently, the therapeutic effect was executed by the PS, while the expression of IL12 in the tumor’s vicinity recruited immune factors, resulting in a robust immune response. This approach effectively overcame tumor immunosuppression, a common characteristic of tumors, and established long-term immunity to prevent tumor recurrence. Furthermore, its target specificity, attributed to its functional properties, localized the IL12 expression at the tumor site, minimizing the cytotoxicity to healthy cells [122,128].

## 6. PDT Applications in 3D Cancer Models

Three-dimensional tumor spheroids are an excellent model system since they consider 3D cell–cell interactions, and the extracellular matrix is like tumors in vivo. The application of hypericin as a PS has revealed promising outcomes as hypericin is capable of penetrating the core of the tissue after 30 min of incubation, triggering O_2_ activation proportional to the dose applied [129].

Recent preclinical studies have also demonstrated the effectiveness of 5-ALA in treating glioblastoma within brain organoids, inducing cancer cell death while sparing healthy cells [130]. While this study focused on exploring the efficacy of 5-ALA to treat glioblastoma, their findings on the selective effectiveness in brain organoids provide a foundation (Figure 4). Additionally, the cholangiocarcinoma (CCA) organoids and monolayer structures of non-tumor organoids established by Fujiwara et al. demonstrated a remarkably high photodynamic activity based on a higher accumulation of protoporphyrin IX as a metabolite of 5-ALA compared to non-tumor organoids (40–71% vs. <4%, respectively), which suggested that 5-ALA-based photodynamic activity had some diagnostic potential for the discrimination of CCA from non-tumor tissues [131].

Finally, Broekgaarden et al. cultured the tumor organoid with metastatic human pancreatic cancer cells AsPC-1 on the solidified matrigel scaffold according to the established 3D adherence culture scheme, ingrained the in vitro micro-metastatic pancreatic cancer model, and investigated the potential of a combined treatment consisting of PDT and subsequent oxaliplatin chemotherapy [132].

The outcomes demonstrated that neoadjuvant PDT enhanced the immediate and prolonged efficacy of oxaliplatin in metastatic pancreatic organoid carcinoma. It follows that the organoid model with its unique characteristics would certainly play a very large role in the investigations of PDT treatments of tumors. From this, one might extrapolate the potential usefulness of organoids as a valuable model for more targeted and physiologically relevant PDT research. Organoids, with their variety of cell types and 3D structure, closely mimic human organ tissues, offering a more realistic environment compared to traditional 2D cell cultures. Their potential similarity in optical properties to actual tissues may allow for a more accurate assessment of light absorption, crucial in PDT. Organoids can be customized to simulate specific tissue characteristics, such as tissue thickness, which influences light absorption and the efficacy of the PS. Moreover, tailoring microenvironmental factors like the extracellular matrix and oxygen levels is crucial, as these elements significantly impact the efficacy and overall behavior of the PS in PDT. These attributes are vital for fine-tuning PDT parameters, including PS selection, light sources, treatment duration, and drug-light intervals. Organoids can provide a dynamic platform for the in-depth exploration of PDT’s mechanisms and interactions, enhancing real-time in vitro monitoring and overall treatment optimization.

## 7. Conclusions

In this review, we compile the latest PDT advances from the last two decades. We emphasize the basic PDT principles from a photophysical–chemical point of view, showing how PS molecules act on the target tissue and what mechanisms they can adopt to produce cytotoxicity. We depict the role of the three essential elements, light, oxygen, and the PS, to improve phototherapeutic results, as well as the progression and alternatives that have been explored in recent years for the conception of improved therapeutic modalities to overcome the PDT weaknesses.

This review presents the results of only some of the studies on PSs and hybrid materials used for PDT in combination with radiotherapy, nanotechnology, nanomaterials, and immunotherapy. Unfortunately, few clinical trials are being conducted to determine the actual impact of the combination of these therapeutic approaches. Therefore, we expect in the next decade more clinical trials confirming the efficacy of combining PDT specifically with immunotherapy, conducted at various centers around the world. Multiple clinical trials could result in new anticancer therapy options being brought to market, which would contribute to the widespread use of PDT in oncology.

## Figures and Tables

**Figure 1 ijms-25-01023-f001:**
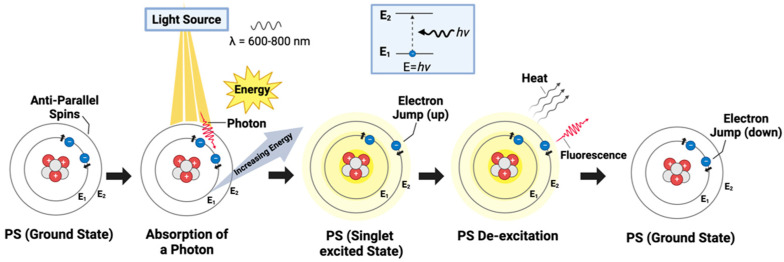
Basic mechanism of photosensitizer excitation in PDT: The photosensitizer (PS) is represented as a single atom for conceptual ease, highlighting the electron transitions between energy levels. The diagram illustrates the initial stage of PS activation, where the PS, in its ground state, absorbs a photon within the therapeutic window (600–800 m), elevating an electron to the singlet excited state. The inset clarifies the electron’s energy jump and subsequent fluorescence emission as it returns to the ground state, a precursor event before intersystem crossing and reactive oxygen species generation in PDT.

**Figure 2 ijms-25-01023-f002:**
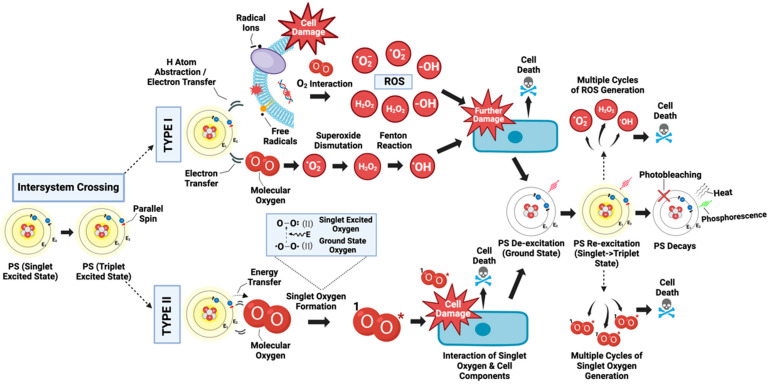
Triplet excited state dynamics in PDT: The diagram captures the photodynamic action of a PS which, upon light absorption, transitions from a ground state to a singlet excited state. It then undergoes intersystem crossing to achieve a triplet excited state. In Type I reactions, the triplet PS engages with cellular substrates to simultaneously generate radical ions or free radicals. These entities can inflict damage on cellular substrates and, through further reactions with molecular oxygen, give rise to diverse ROS that cause enhanced damage. Concurrently, the PS can directly interact with oxygen to initiate a reaction cascade, leading to potent damage from the hydroxyl radical, resulting in cell death. In Type II reactions, the triplet PS directly transfers energy to molecular oxygen, forming singlet oxygen (^1^O_2_) that damages cellular components. Both Type I and Type II pathways can proceed in tandem, with repeated cycles of ROS generation culminating in extensive cell damage, cell death, and the degradation of the PS.

**Figure 3 ijms-25-01023-f003:**
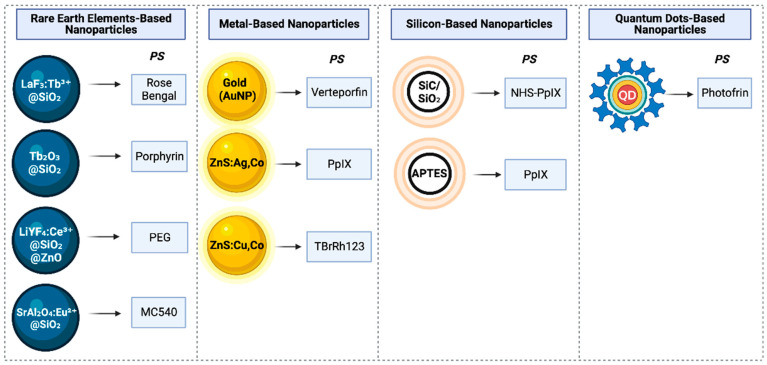
Simplified overview of nanoparticle classifications in X-ray photodynamic therapy (X-PDT). The diagram provides a streamlined depiction of the four major nanoparticle categories, distinguished by scintillator composition, and matched with their corresponding photosensitizers (PSs). Categories include rare earth element-based, metal-based, silicon-based, and quantum dot-based nanoparticles, associated with PSs like Rose Bengal, Verteporfin, NHS-PpIX, and Photofrin, respectively.

**Figure 4 ijms-25-01023-f004:**
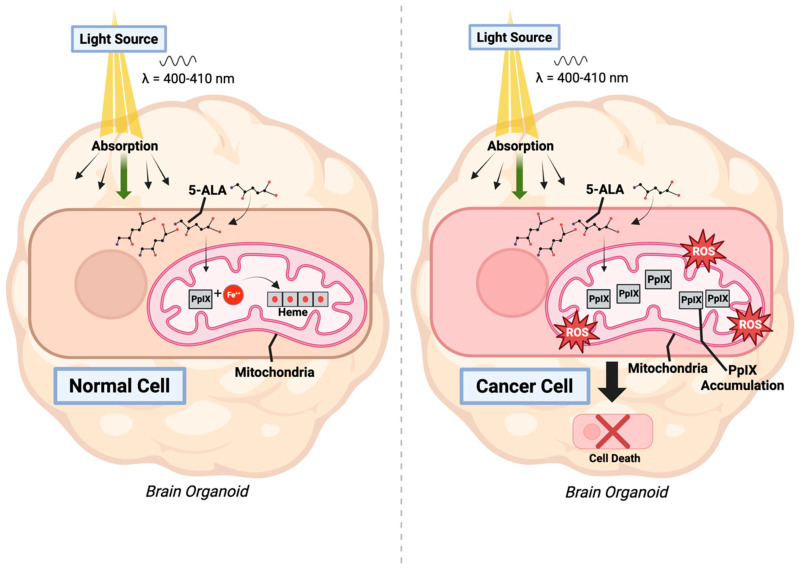
Photodynamic therapy in Induced Pluripotent Stem Cell (iPSC)-derived organoids co-cultured with tumor cells.

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
