# Peer review of "Current Advances in Photodynamic Therapy (PDT) and the Future Potential of PDT-Combinatorial Cancer Therapies"

_ijms, 2024, doi:10.3390/ijms25021023_

Round 1

Reviewer 1 Report

Comments and Suggestions for Authors

This paper summarized the work on PDT advances from the last two decades, which presents the results of some of the studies on photosensitizers and hybrid materials used for PDT in combination with radiotherapy, nanotechnology, nano-materials, and immunotherapy. I think it could be a publishable work after major revision. However, the current version cannot be published. Below are some suggestions for the author to revise the paper.

1. Some parts of introduction are too basic, such as sections 2-4. Please simplify them.

2. Instead of listing all the works equally, the authors should emphasize on the more important ones and latest developments. For example, on page 9 line 429, elevating photodynamic therapy´s efficacy in hypoxic tumors is a very promising research direction. The authors should do a detailed understanding of this direction and summarize the advantages and disadvantages of each approach.

3. On page 10 line 488, what is the mechanism of reactive oxygen generation by quantum dots, and what is the difference between it and general photosensitizers? Please explain it in more details, including the advantages and disadvantages of quantum dots-PDT.

4. The arrangement of the contents of this paper lacks logic. The authors should use large and small headings to differentiate contents, not list them equally from section 1to section 14.

5. Please add more illustrations to show the latest developments, such as what kind of PSs are required for X-PDT.

6. The superscripts and subscripts of many species are not correct, such as “singlet oxygen, oxygen, superoxide anion, iron and etc.” in lines 209-232 on page 5.

Comments on the Quality of English Language

The superscripts and subscripts of many species are not correct, such as “singlet oxygen, oxygen, superoxide anion, iron and etc.” in lines 209-232 on page 5.

Author Response

This paper summarized the work on PDT advances from the last two decades, which presents the results of some of the studies on photosensitizers and hybrid materials used for PDT in combination with radiotherapy, nanotechnology, nano-materials, and immunotherapy. I think it could be a publishable work after major revision. However, the current version cannot be published. Below are some suggestions for the author to revise the paper.

  1. Some parts of introduction are too basic, such as sections 2-4. Please simplify them.

We agree with this reviewer comment and we have deleted section 2 and 3 from the previous version as it was very little informative and repetitive with other sections. Additionally,  we have extended those sections comenting the newest advances in the field that could be more interesting.

  1. Instead of listing all the works equally, the authors should emphasize on the more important ones and latest developments. For example, on page 9 line 429, elevating photodynamic therapy´s efficacy in hypoxic tumors is a very promising research direction. The authors should do a detailed understanding of this direction and summarize the advantages and disadvantages of each approach.

We agree with this reviewer comment, that the previous version had too many sections with no hierarchy. The updated version has better organized the respective sections. We have made a big section where we coment the challenges in PDT. Within this section we explain the advances or strategies followed to enhance the PDT’s efficacy in Hipoxic tumors, X-Ray induced Photodynamic Therapy (X-PDT), the aplication of quantum dots in PDT as well as the immunological strategies in PDT.

Additionally, as PDT’s efficacy in hypoxic tumors is a very interesting challenge to achieve, we have significantly extended this section. We mention now more strategies such as delivering O2 to the tumors, in situ generation of O2 and decreasing the O2 consuption during PDT by the design of Type I photosensitizers.

The new parragraph stands as:

To address this problem, researchers have been developing advanced nanoplatforms and strategies to enhance the therapeutic effect of PDT in tumor treatment. Such strategies include delivering O2 to the tumors, in situ generation of O2; and decreasing the O2 consumption during PDT by the design of Type I photosensitizers. As O2 carriers, some nanomaterials have been developed such as hemoglobin (Hb), red blood cell (RBC), perfluorocarbons (PFC), and metal-organic frameworks (MOFs) [64,65]. Alternatively, in situ O2 generation through bio-chemical reactions between the components in drug carriers and the endogenous substances in tumor tissues have also been explored. Among them, the use of H2O2 as oxygen-containing molecule to relieve tumor hypoxia has been widely confirmed[66]. To date, catalase (CAT) and CAT-like enzyme are applied to catalyze the decomposition of H2O2 into water (H2O) and O2, to relieve tumor hypoxia [67–71]. Although in this case O2 production is also limited due to the limited H2O2 content in the tumor.

And at the end of the section we also have expanded the section including new strategies that combined PDT with other therapeutic methods such as chemotherapy, photothermal therapy, immunotherapy and gas therapy to address the mono PDT in hypoxic environments comenting the advantages and disadvantages of each combination. Please see the updated section.

5.1. Elevating Photodynamic Therapy´s Efficacy in Hypoxic Tumors

  1. On page 10 line 488, what is the mechanism of reactive oxygen generation by quantum dots, and what is the difference between it and general photosensitizers? Please explain it in more details, including the advantages and disadvantages of quantum dots-PDT.

We appreciate this comment from the reviewer and in the updated version of the review we have given more details about the mechanism of reactive oxygen generation by quantum dots.

The following parragraph stands as:

Additionally, the generation of ROS by quantum dots (QDs) is a distinctive process that sets them apart from conventional PSs. This process is primarily driven by the QDs’ unique electronic structure and surface states. When these QDs are exposed to light, they can efficiently transfer energy or electrons to nearby oxygen molecules, leading to the formation of various ROS types like singlet oxygen, superoxide anion, and hydroxyl radicals. The diversity in ROS production is linked to the QDs’ size, shape, and surface chemistry, which influence the potential pathways in their excited state.

QDs differ from traditional photosensitizers in their ability to absorb a broader spectrum of light wavelengths. This broad absorption is attributed to their tunable electronic structures, allowing more effective energy transfers to oxygen molecules and the generation of diverse ROS types. In contrast, conventional PSs usually have limited absorption ranges and may produce fewer ROS types. Moreover, QDs can be engineered for specific targeting, enhancing their effectiveness, particularly in PDT applications [115,116].

  1. The arrangement of the contents of this paper lacks logic. The authors should use large and small headings to differentiate contents, not list them equally from section 1to section 14.

We agree with the comments of this reviewer and in the updated version of the review we have rearranged the sections in bigger blocks. We have grouped section 5 that address the challenges in PDT in 4 subsections that include (1) PDT efficacy in hypoxic tumors (2) X-Ray induced Photodynamic Therapy for improved penetration and ROS generation (3) Enhancing light emision and intensity with quantum dots in PDT (4) Harnessing Immunological Strategies in Photodynamic Therapy.

  1. Please add more illustrations to show the latest developments, such as what kind of PSs are required for X-PDT.

We appreciate very much the suggestion of adding more ilustrations to the review and accordingly we have added a new Figure 4 where we ilustrate the kind of PS that are required for X-PDT.

Figure 4 legend stands as:

Figure 4: Simplified Overview of Nanoparticle Classifications in X-Ray Photodynamic Therapy (X-PDT). The diagram provides a streamlined depiction of the four major nanoparticle categories, distinguished by scintillator composition and matched with their corresponding photosensitizers (PS). Categories include rare earth elements-based, metal-based, silicon-based, and quantum dots-based nanoparticles, associated with photosensitizers like Rose Bengal, Verteporfin, NHS-PpIX, and Photofrin, respectively.

  1. The superscripts and subscripts of many species are not correct, such as “singlet oxygen, oxygen, superoxide anion, iron and etc.” in lines 209-232 on page 5.

We apologize for this mistake in the nomenclature, the updated revision has been revised and corrected accordingly. We appreciate this comment very much.

Reviewer 2 Report

Comments and Suggestions for Authors

In this work, Niuska Alvarez and Ana Sevilla reviewed the topic "Current Advances in Photodynamic Therapy (PDT) and the Future Potential of PDT-Combinatorial Cancer Therapies". Photodynamic therapy is a significant field that can provide the potential for clinical transfer in the future. The authors managed the skeleton well, they started from the foundations of PDT, and explained the details in multiple details and the mechanisms behind it. Besides, they have clearly illustrated the challenges and potential solutions as well as the future directions of PDT.

 Thus, I suggest accepting the paper as it is.

Author Response

We appreciate very much that you liked our work and we hope you also like the revised version uploaded after reviers comments.

Reviewer 3 Report

Comments and Suggestions for Authors

In their paper entitled:“ Current advances in Photodynamic Therapy (PDT) and the future potential of PDT-Combinatorial Cancer Therapies“, the authors have summarised the current state of knowledge about photodynamic therapy and its use in combination treatment as well as promising aspects of the use of organoids in the studies.

The manuscript contains basic information in 10 sections. The novelty begins with  the 11th section, which deals with the use of quantum dots in PDT. A combination with X-rays and a treatment approach is presented.

I have a few comments on this study:

Switching of autophagy and apoptosis in PDT is possible, as has also been shown recently with photobiomodulation and its combination with PDT. This should be discussed (e.g. in section 8, doi: 10.1016/j.jphotobiol.2022.112539).

Section 2 is a very simple explanation of the principles that are better described in the following sections. I suggest introducing them in this section. E.g. photosensitisers of different generations in the section in lines 62-70, type I and II reaction in the section in lines 76-85.

The approach of investigating spheroids as a model in PDT is missing (doi: 10.1007/s00216-022-04107-2).

iPSC in section 14 as an abbreviation should be explained.

The text in Figure 3 is too small and may not be legible.

Author Response

In their paper entitled:“ Current advances in Photodynamic Therapy (PDT) and the future potential of PDT-Combinatorial Cancer Therapies“, the authors have summarised the current state of knowledge about photodynamic therapy and its use in combination treatment as well as promising aspects of the use of organoids in the studies.

The manuscript contains basic information in 10 sections. The novelty begins with  the 11th section, which deals with the use of quantum dots in PDT. A combination with X-rays and a treatment approach is presented.

We agree with the comment of this reviewer and according to that, we have deleted section 2 and 3 for beeing too simple and reestructured and expandedt information in other sections that could be more interesting as they are newer in the field.

I have a few comments on this study:

1.-Switching of autophagy and apoptosis in PDT is possible, as has also been shown recently with photobiomodulation and its combination with PDT. This should be discussed (e.g. in section 8, doi: 10.1016/j.jphotobiol.2022.112539).

We appreciate very much the suggestions of this reviewer,  therefore, we have commented this work in our review.

Please see the added text to our updated review.

On the other hand, other studies have suggested that autophagy can promote a pro survival effect as an independent mechanism of cell death, separate from apoptosis [50]. In this line, recent studies have described that even though autophagic proteins have been shown to be upregulated by hypericin-induced combination treatment of photobiomodulation and PDT in human dermal fibroblasts (HDF), autofagosome degradation was inhibited in this HDF. Therefore, autophagy seems to have a pro survival effect in HDF under this treatment but not in U87 glioblastoma cells, thus, giving a selective cell death toxicity [57].

.”

2.-Section 2 is a very simple explanation of the principles that are better described in the following sections. I suggest introducing them in this section. E.g. photosensitisers of different generations in the section in lines 62-70, type I and II reaction in the section in lines 76-85.

We agree with the comment of this reviewer and accordingly we have deleted section 2 and 3 as they were not giving too much information and the same concepts were better explained in the following sections such as the diferent generations of photosensitizers and the Type I and II reactions. This also has given us space to extend other more novel tòpics that could be more interesting and atractive to the reader.

3.-The approach of investigating spheroids as a model in PDT is missing (doi: 10.1007/s00216-022-04107-2).

We appreciate this comment, and in the new version of the updated review we have added this work, as well as others, within section 6 entittled. PDT Applications in 3D cancer models.

The added text stands as:

3D tumor spheroids are an excellent model system since they consider 3D cell-cell interactions, and the extracellular matrix is like tumors in vivo. The application of hypericin as photosensitizer has revealed promising outcomes as hypericin is capable to penetrate the core of the tissue after 30 min incubation triggering O2 activation proportional to the dose applied [99].”

And we have also added another work using organoids and 5-ALA treatment for the cholangiocarcinoma.

‘Additionally, the cholangiocarcinoma (CCA) organoids and monolayer structures of non-tumor organoids established by Fujiwara et al demonstrated a remarkably high photodynamic activity based on higher accumulation of protoporphyrin IX as a metabolite of 5-ALA compared to non-tumor organoids (40–71% vs. < 4%, respectively), which suggested that 5-ALA-based photodynamic activity had some diagnostic potential for the discrimination of CCA from non-tumor tissues [131].”

4.-iPSC in section 14 as an abbreviation should be explained.

We apologize for not having especify the meaning of the induced pluripotent stem cells (iPSC) in the revised version of the revised review we have explained this abbreviation.

Figure 4 legend

Figure 4. Photodynamic therapy in Induced Pluripotent Stem Cells (iPSC) derived organoids co-cultured with tumor cells.

5.-The text in Figure 3 is too small and may not be legible.

We agree with the comments of this reviewer about the font size in Figure 3, therefore we have increased  the font size in all the Figures of the Review.

Round 2

Reviewer 1 Report

Comments and Suggestions for Authors

The Authors have considered and addressed all my comments, in particular about the content--elevating photodynamic therapy´s efficacy in hypoxic tumors. I think this revised manuscript can be accepted for publication.

Comments on the Quality of English Language

None